# Rapid Food Authentication Using a Portable Laser-Induced Breakdown Spectroscopy System

**DOI:** 10.3390/foods12020402

**Published:** 2023-01-14

**Authors:** Xi Wu, Sungho Shin, Carmen Gondhalekar, Valery Patsekin, Euiwon Bae, J. Paul Robinson, Bartek Rajwa

**Affiliations:** 1Department of Basic Medical Sciences, Purdue University, West Lafayette, IN 47907, USA; 2Weldon School of Biomedical Engineering, Purdue University, West Lafayette, IN 47907, USA; 3School of Mechanical Engineering, Purdue University, West Lafayette, IN 47907, USA; 4Bindley Bioscience Center, Purdue University, West Lafayette, IN 47907, USA

**Keywords:** authentication, LIBS, spectroscopy, food fraud

## Abstract

Laser-induced breakdown spectroscopy (LIBS) is an atomic-emission spectroscopy technique that employs a focused laser beam to produce microplasma. Although LIBS was designed for applications in the field of materials science, it has lately been proposed as a method for the compositional analysis of agricultural goods. We deployed commercial handheld LIBS equipment to illustrate the performance of this promising optical technology in the context of food authentication, as the growing incidence of food fraud necessitates the development of novel portable methods for detection. We focused on regional agricultural commodities such as European Alpine-style cheeses, coffee, spices, balsamic vinegar, and vanilla extracts. Liquid examples, including seven balsamic vinegar products and six representatives of vanilla extract, were measured on a nitrocellulose membrane. No sample preparation was required for solid foods, which consisted of seven brands of coffee beans, sixteen varieties of Alpine-style cheeses, and eight different spices. The pre-processed and standardized LIBS spectra were used to train and test the elastic net-regularized multinomial classifier. The performance of the portable and benchtop LIBS systems was compared and described. The results indicate that field-deployable, portable LIBS devices provide a robust, accurate, and simple-to-use platform for agricultural product verification that requires minimal sample preparation, if any.

## 1. Introduction

Food fraud, including economically motivated adulteration (EMA), is defined by the US Food and Drug Administration (FDA) as an act in which a valuable ingredient or component of a food product is intentionally omitted, removed, or replaced by a substitute. EMA occurs, as well, when a substance is added to food in order to enhance its appearance, taste, or perceived value [1,2,3]. Food fraud may involve the deliberate and intentional substitution, addition, tampering, or misrepresentation of food, food ingredients, qualities, or food packaging [2,4].

According to the Food Fraud Database (Decernis LLC, Washington, DC, USA), common examples of affected foods include coffee, cheese, olive oil, herbs and spices, seafood, meat, poultry, alcoholic beverages, honey, fruit and vegetable juices, and cereals. As of 2017, the greatest number of food fraud incidents was associated with dairy products [5,6,7]. The quality of dairy products in general, and cheeses in particular, was the most frequently reported issue in terms of safety (presence of pathogenic microorganisms), fraud incidences (fraudulent documentation), and adulteration (presence of wood pulp) [7,8,9,10,11]. Many highly valued artisanal cheeses are identified by protected designation of origin (PDO), which helps protect small manufacturers (and local economies) by guaranteeing the authenticity of their products and supporting quality maintenance [12]. Hence, in this study, we selected European Alpine-style cheeses, in addition to coffee, powdered spices, vanilla extract, and balsamic vinegar, to demonstrate the efficacy of our approach [13,14,15,16]. A rapidly growing number of reports on food fraud further emphasize the importance of the topic [17].

Rapid classification and authentication of food ensure that fraudulent products do not reach the market or are quickly and efficiently withdrawn. Vibrational spectroscopy, fluorescence spectroscopy, hyperspectral imaging, PCR-based approaches, mass spectrometry, and liquid chromatography are the currently used technologies for detecting food adulterants specifically and food fraud in general [18,19,20,21,22,23]. Regrettably, each of these approaches requires extensive sample preparation, costly laboratory equipment, highly skilled technicians, and, in some instances, multiple chemical reagents. Regardless of which method is used, there is a considerable time factor associated with the analytical steps.

Laser-induced breakdown spectroscopy (LIBS) has previously been explored as an analytical approach for assessing food integrity [22,24,25,26,27,28,29,30], and it is considered to be a promising and exciting method by experts [28,31]. It is a technique that directs a high-energy laser pulse to the surface of a material, resulting in the generation of plasma above this surface and the subsequent emission of optical radiation characteristic of the elements, ions, and molecules that originally comprised the sample [28,32,33]. Analyses of the plasma’s optical emission can be used to determine the elemental makeup of the source material [34]. The advantages of LIBS include multi-element detection ability, speed of sampling, and compatibility with a variety of samples (solids, liquids, and gases) [22,33]. In addition, LIBS requires minimal sample preparation and can be used in tandem with other analytical techniques, such as mass spectrometry and Raman spectroscopy [35,36]. LIBS has been used to evaluate milk, infant formula, butter, honey, bakery products, coffee, tea, vegetable oils, water, cereals, flour, potatoes, palm dates, and various types of meat [27,34,37,38,39,40,41,42,43,44,45,46,47,48,49]. Moncayo et al. [50] employed LIBS for the authentication of red wines and the localization of their geographic origin. Bilge, et al. [45] discriminated between beef, chicken, and pork meats using LIBS. LIBS was used to identify kudzu powder from different habitats [51], establish the geographical origin of rice [24,52,53], and identify olive oil [54,55,56].

Herein, the purpose of this study was to determine whether LIBS was a viable choice for identifying food products in various forms (liquid, solid, and powder food samples), using classification models to detect food fraud cases (mislabeling). Two LIBS systems were evaluated to establish the analytical capabilities of LIBS: a benchtop laboratory-based system and a portable device. To our knowledge, this is the first study to use portable LIBS systems for classification analysis of these high-value food goods with the goal of ensuring their authenticity. This is critical since the long-term efficacy of LIBS-based food authentication depends on the availability of portable diagnostic equipment capable of preventing food fraud across the commercial distribution chain, especially for highly valued commodities.

## 2. Materials and Methods

### 2.1. Types of Food Samples and Sample Preparation

LIBS is often used on solid samples like metal and plastic that can be recycled. However, food samples in general and liquid food samples in particular present some extra challenges. Because of this, we chose several types of food samples, including liquids, solids, and powders, to represent a wide range of product categories (Table 1). 

#### 2.1.1. Liquid Samples

##### Balsamic Vinegar

Six types of balsamic vinegar were acquired and tested in the study. These examples were chosen to represent the major brands with distinct protected designations of origin, including three different brands of Modena balsamic vinegar from Italy, barrel-aged balsamic vinegar from Napa Valley (Nap, CA, USA), and Gran Deposito Aceto Balsamico di Modena (Italy), as well as a sample of home-produced barrel-aged balsamic vinegar generously provided by Prof. Andrea Cossarizza (the University of Modena and Reggio Emilia, Italy). A list of the brand names of balsamic vinegar used in the study is provided in Table A1 in Appendix A.

For the measurements of liquid samples in the study, a method utilizing nitrocellulose paper was used. Ten microliters of a sample were deposited onto a 6 × 6-mm nitrocellulose square. Four independent samples of each product were analyzed. There was uneven sample distribution exhibited on the nitrocellulose paper from two samples due to their viscosity. One-to-one dilution with deionized water (DI) was used to resolve it. Samples containing only 10 μL of MilliQ on nitrocellulose squares were used as negative controls. Each nitrocellulose square was measured at different locations 25 times to account for variability and augment the representative dataset.

##### Vanilla Extracts

A total of six vanilla extract samples were acquired for this study from local stores (West Lafayette, IN). Among them were four vanilla extracts from different geographic locations, represented by different brands, and one vanilla syrup; the remaining one was an imitation vanilla extract composed using artificial flavors. Brand names of the six vanilla products measured in the study are listed in Table A2, Appendix A.

A method similar to that used for measuring the balsamic vinegar (nitrocellulose) was employed for the vanilla extract samples. Briefly, 10 μL of each sample was deposited on a 6 × 6-mm nitrocellulose square and dried at room temperature for 30 min. Each brand was represented by four nitrocellulose-based samples. Due to the high viscosity of the vanilla syrup, one-to-two dilutions with DI water were prepared. As before, 10 μL of DI water on nitrocellulose squares served as the negative control. Each nitrocellulose square was shot 25 times at multiple locations. 

#### 2.1.2. Solid Samples

##### Cheeses

Fifteen types of European Alpine-style cheese purchased from iGourmet, a web-based food delivery service, were shipped as refrigerated 5- to 10-oz. blocks (from 141.75 to 283.5 g). Separately, American Gruyère-style cheese was purchased from a local Kroger supermarket. This product is referred to as Wisconsin Gruyère cheese in the study. A total of 16 types of cheeses are listed in Table A3, Appendix A.

Cheeses were stored at 4 ± 1 °C until analysis. Approximately 1 cm of the outside of the cheese block was cut and discarded to prevent the use of dried material. For LIBS measurement, cheese samples were cut into rectangular slices of uniform thickness (approximately 10 mm wide, 10 mm long, and 2 mm thick) using a stainless-steel blade. For each time point, four replicate specimens were cut from each type of cheese block. The blade was rinsed and cleaned with ethanol and dried between each cut of the same cheese and between each cut of different cheeses. 

Water activity (a_w_) was determined for the sixteen Alpine-style cheeses every two weeks for 42 days of storage in a refrigerator. The purpose was to establish data regarding the impact of storage on the LIBS-based product classification. In short, grated cheese samples (0.5 g) were placed in plastic dishes, covered, stored at 4 °C, and assayed in duplicate at 25 °C on an AquaLab 4TE Dew Point Water Activity Meter (AquaLab, Pullman, WA, USA). The precise dewpoint temperature of the sample was established by an infrared beam focused on a small mirror. The temperature at the dewpoint was then converted into water activity. Prior to analysis, the machine was calibrated using a certified AQUA LAB standard (Lot no. 20805392, 0.920 a_w_ NaCl, 2.33 mol/Kg in H_2_O). The a_w_ of the cheese was measured at 0 (T1), 14 (T2), 28 (T3), and 42 (T4) days, along with the LIBS measurement. The a_w_ data were expressed as the mean of three repetitions in three independent measurements. Utilizing commercially accessible software, data were analyzed using 2-way ANOVA and Tukey’s multiple comparisons test (OriginPro, OriginLab Corporation, Northampton, MA, USA). Comparisons were considered significantly different at a *p*-value < 0.05.

##### Coffee Beans

In this study, seven varieties of coffee were tested directly without the need for grinding or milling. Whole coffee beans were stored in the original sealed package until the test and resealed after use. The names of the coffee varieties tested in the study are listed in Table A4, Appendix A.

Four randomly selected coffee beans of each type were measured from both the front and back sides. To avoid additional variability caused by the movement of the beans when hit by the laser, the coffee beans were fastened with tape to a sample holder. The location of the beans was adjusted for multiple LIBS interrogations to cover as much area on the bean surface as possible. 

#### 2.1.3. Powdered Food Samples

##### Spices

Six different types of spices were chosen and purchased from the retail outlets. Table A5 in Appendix A provides the brand names of the spices evaluated in the study.

Most of the ground spices used in this study are fine powders, although the classic nutmeg is roughly milled powder. The red pepper comes as flakes, which splash easily when hit by laser shots. Therefore, we employed a sample holder when performing the measurements.

### 2.2. Benchtop and Handheld LIBS Systems Setup

The custom-built benchtop LIBS system is shown in Figure 1a and consists of a Nano SG 150-10 pulsed Nd:YAG laser (Litron Lasers, Bozeman, MT, USA). The laser had a pulse width of 4 ns; a pulse energy of 62 mJ was used in this study. The ablation laser’s spot size was approximately 700 µm. Details on the optics used to direct the alignment and the ablation laser beams were described previously [57,58]. Emissions were detected by an AvaSpec-Mini-VIS-OEM spectrometer (Avantes, Apeldoorn, the Netherlands), which has a 350–600-nm spectral range with 0.33-nm resolution. Target samples were placed on a motorized XYZ stage. The stage height was adjusted so that the crosshairs of the two lasers assisting in sample positioning were visible at the surface of the samples. A digital delay pulse generator controlled the triggering of the ablation laser, motorized stage, and spectrometer. The delay between the ablation pulse and spectrometer data acquisition was 1.17 µs. 

The Z-300 LIBS Analyzer (SciAps, Inc., Boston, MA, USA) is a commercially available handheld LIBS system. The laser, spectrometer, optics, argon gas cartridge, electronics, and control module were housed in a gun-shaped enclosure, as illustrated in Figure 1b. Measurements were performed when the sample window (3 cm by 3 cm) was covered with samples, followed by laser activation. The LIBS analyzer uses a pulsed laser, 5–6 mJ/pulse, and 1- to 2-ns pulse width. The laser spot size was 100 µm. The spectral range was approximately 190–950 nm. The settings for rastering location and repetition rate were controlled in the Profile Builder software (SciAps, Inc.) as needed.

All measurements were taken at 25 different locations across a 5 × 5 rastering array of four different specimens representing each individual food product. The measurements of cheeses were repeated at multiple time points (Figure 2). Each spot was ablated with a single laser shot. Accordingly, 100 spectra per food type per time point were analyzed for classification. LIBS measurements were performed using both benchtop and handheld systems for each type of food sample involved in the study. 

### 2.3. Classification Procedures 

Raw spectra were filtered to eliminate low signal-to-noise instances due to faulty sample positioning or similar technical problems. Spectral normalization and a median filter were applied to reduce the effects of variations in emission intensity coming from plasma fluctuations. Subsequently, every spectral feature was used in multiple ANOVA models as a dependent variable in order to select the features associated with large effect sizes (η^2^) [59]. This was followed by the training of a regularized multinomial logistic regression elastic net model (ENET), which provides multivariate feature selection as well as classification (prediction) [60,61]. ENET combines LASSO and ridge regression techniques. Although the use of the ENET approach in LIBS data analysis has been reported before [62], despite its advantages, it is still a very uncommon method for this field, which traditionally relies on well-established chemometric techniques such as PLS-DA [63,64,65,66]. Importantly, in the n≪p setting, it retains the sparse features of LASSO regression and the stability of ridge regression. Note that the number of selected features per food type could differ for each ENET model. The ENET prediction quality was evaluated using 10-fold cross-validation.

## 3. Results

### 3.1. LIBS Measurements

Table 1 summarizes all the food products measured in the study. We tested three different forms of high-value regional foods (liquid, solid/semi-solid, and powder) by both benchtop and handheld LIBS, including 16 hard cheeses, seven coffee varieties, six vanilla/vanillin extracts, and six different powdered spices. Additionally, we monitored changes in the water activity of the cheese samples at four sampling time points. It is known that water-activity measurement is an important method for predicting the shelf life of food products. By measuring and controlling the water activity of foodstuffs, it is possible to monitor and maintain the physical stability of foods and optimize their physical properties. Therefore, the water activity of cheeses is an indicator informing us about the shelf-life status of the product. Figure 3 illustrates the evolution of water activity in the test cheeses during a period of refrigerated storage. 

All the food samples were measured by the benchtop LIBS system covering a spectral window from 200 to 600 nm. The corresponding data obtained from the handheld LIBS device covered a spectral range of 190 to 950 nm. The typical LIBS spectra of (a) coffee bean, (b) vanilla extract, (c) balsamic vinegar, and (d) spice samples, measured using benchtop and handheld LIBS systems, are shown in Figure 4 and Figure 5, respectively. The spectra of each food category represent an average of all the measurements. For example, Figure 4b is an averaged spectrum of 600 (six vanilla extracts × 100 spectra/vanilla extract) measurements. The data in Figure 5 are spectral results obtained after automatic data processing executed in the handheld device, whereas Figure 4 represents the raw data from the benchtop system. The main emission lines from the essential elements for food analysis, selected as inputs of ENET, have been labeled in Figure 4a and Figure 5a. The detected elemental emission bands are identified with the aid of the spectroscopic data included in the NIST Atomic Spectra Database [67]. CN band, Ca ionic, Ca atomic, C_2_ band, P ionic, and Na atomic peaks, which are dominantly detected in biomaterials, can be seen in Figure 4.

Although there was a minor difference in peak values depending on the food products, the same emission peaks were found in all the tested food samples. Similarly, there were only minor differences in the handheld LIBS results, as shown in Figure 5. However, additional peaks, such as C, Mg, H, K, and O peaks, were detected owing to the broader spectral range (190–950 nm) of the handheld device. This broader spectral range contributed to improving the classification accuracy of the coffee bean, vanilla extract, and balsamic vinegar samples.

Figure 6 and Figure 7 show the averaged LIBS spectra of the cheese samples, measured using the benchtop and handheld LIBS systems at four different time points. Note that each spectrum is an average of 1600 (16 cheese types × 100 spectra/cheese type) measurements under the same conditions. The measurements were conducted every 14 days. The cheese specimens were instantly stored in a vacuum pack and refrigerator after each measurement. Emissions of the identical elemental components in various LIBS spectral fingerprints of the cheese samples led to a significant degree of resemblance. Some minor differences in peak intensities appeared at different time points. As an example of changes over time, Table 2 compares the integrated peak intensity of Na I 589.0 nm in Frantal Emmental Cheese (C10) at each sampling time point. Integrated peak intensity was calculated by integrating the peak area study after sum-to-one normalization. It was shown that the averaged integrated intensities of the Na I emission peak were similar at four different sampling time points, implying relatively uniform product preservation within time periods. 

### 3.2. Classification Using the Elastic Net Approach

Table 3 reports the ENET classification accuracy of five different food products measured using the benchtop LIBS system and the handheld LIBS system. The training (and accuracy evaluation) was performed via 10-fold cross-validation. As can be seen in the tables, cheese samples were sampled and measured by two LIBS systems at four time points. Thus, separate classifiers were developed and applied to the dataset containing measurements from each of the four time points. As mentioned before, prior to the algorithmic training, univariate feature selection via ANOVA was applied to the data to minimize the subsequent training time. The accuracy of the model was found to be excellent, reaching 94.5 ± 1.51% for vanilla extract and 99.30 ± 0.70% for spices in the benchtop system, and 92.70 ± 2.30% for coffee beans, 98.30 ± 0.69% for vanilla extract, and 90.80 ± 1.88% for balsamic vinegar in the handheld system.

The classification of coffees and balsamic vinegar showed slightly lower accuracy in the benchtop system compared to the handheld system. This suggests that the broad spectral range of the handheld system may be the most dominant factor in the classification of coffee beans and balsamic vinegar using LIBS. However, the classification accuracy of spices in powder form was lower using the handheld system, pointing to the spectral resolution as the decisive factor. Additional studies are required to evaluate these types of samples further, particularly with respect to the preparation methods for powders. The test results for vanilla extracts show comparable classification accuracy in both LIBS systems. 

The classification performance for cheese samples measured at different storage time points was also assessed. There were no observable differences in the measurements obtained during different periods. The classification accuracy of those measurements did not present significant differences either. Note that every three sample replicates were averaged and analyzed to establish the classification performance results. Slightly higher classification accuracy of cheese samples was shown in the benchtop system than in the handheld device. It is likely that more sensitive detection in the visible and near-visible range (350–650-nm wavelength) could be the critical factor for classifying cheeses using LIBS. 

### 3.3. Food Fraud Detection

In the final step of our study, we simulated two realistic food-fraud detection scenarios in which a specific sample with a different origin and/or composition than the rest of the set was to be identified and detected. In the first simulation, we aimed to identify Wisconsin Gruyère-style cheese manufactured in the USA from pasteurized milk. In the second scenario, we attempted to identify imitation vanilla taste (vanillin) among natural vanilla extracts. In the first scenario, we envisioned three classes (unpasteurized European cheeses branded as “Gruyère” vs. other unpasteurized European Alpine-style cheeses vs. Wisconsin Gruyère-type cheese produced from pasteurized milk), whereas, in the second scenario, there are only two classes (real vanilla extract vs. imitation vanilla flavor). We used multiple repeated independent instances of 5 × 2 cross-validation runs to evaluate the system. For the cheese detection scenario, the accuracies of the benchtop (90.17 ± 1.04%) and the portable platforms (90.95 ± 1.05%) were virtually identical (see Table A6). Similarly, the benchtop and the portable systems operated equally well in detecting the imitation vanilla (99.66 ± 0.47 and 99.38 ± 0.58%, respectively). See Table A7 in Appendix A for the result of the individual classification runs.

## 4. Discussion

### 4.1. Sample Preparation

Solid specimens were successfully analyzed without any processing. Grinding samples into powder and pressing them into a pellet is a popular preparation method for solid foods [68,69]. For instance, Iqbal et al. [70] reported that samples were finely powdered and vacuum-dried at 370 K for 10 h. The sample was then compressed for 20 min at 30 T hydraulic pressure into pellets that were 3 mm thick and 1.3 cm in diameter. However, the preparation of pellets or tablets is an important limiting factor and cannot be easily used for in-situ analysis. In contrast, in our experiments, solid food samples like Alpine cheeses and coffee were tested without any preparation. The samples were immobilized for an easy location adjustment to ensure coverage of the whole sample surface by laser shots during the collection of complete elemental profiles. 

Regarding measurement preparation for powders (spices), we utilized a custom sample holder to confine the samples. To overcome blowing off and scattering during laser-matter interaction, a layer of powdered material was applied to a double-sided piece of tape that covered and adhered to the bottom of the sample holder.

To prevent splashing and the formation of surface ripples caused by the shock wave of LIBS, as well as to achieve a lower limit of detection, better repeatability, and greater sensitivity when working with liquid food samples, the formation of a gel using commercial collagen is commonly performed, followed by drying in an air-assisted oven [50,71,72]. However, the dry gel emission signals cannot be simply subtracted, and additional chemometric spectral treatments are necessary. In our study, we employed a nitrocellulose paper-based sample-preparation approach that is highly compatible with liquid food samples owing to its porous structure, hydrophilic property, and minimal effect on the sample spectra. This approach has been successfully used by other researchers when utilizing LIBS to measure the presence of metals in water or oil [73,74,75,76]. Moreover, this method is simpler and more efficient than the commonly used gel-formation technique [77]. The characteristic peaks of the nitrocellulose membrane do not interfere with the elemental profiles of foods and can be easily distinguished from the LIBS spectral matrix. This is the first report on the use of nitrocellulose membranes with LIBS for the classification of liquid food samples. Compared to the commonly used methods, our approach requires little or no sample preparation. It is simple, rapid, and cost-effective. Consequently, it is more practical and compatible with envisioned usage scenarios involving wholesalers, food inspectors, and customs officers that examine traded agricultural products. However, we must stress that the viability of using nitrocellulose paper may depend on the viscosity of the sample. We have not tested a sufficient range of liquid products to endorse this method unreservedly.

### 4.2. Water Activity

Most of the 16 types of cheese showed a small but statistically significant difference in water-activity values between the beginning of storage and 42 days later. However, despite these small changes in water activity, the classification of cheeses with LIBS systems remained stable and robust. Interestingly, one recent LIBS application was to measure the moisture content in cheese, using oxygen emission normalized by CN emission as the indicator [78]. Another study performed by Ayvaz et al. [79] investigated the potential of using LIBS with partial least squares regression to determine the chemical quality-control parameters for cheese samples, such as moisture, dry matter, salt, total ash, total protein, and pH. In general, our results indicate that small variations in a_w_ are unlikely to be limiting factors for the use of LIBS in authentication, provided that the classification system is paired with an appropriate feature-selection strategy.

### 4.3. Spectral Classification

As anticipated, the LIBS spectra of all the analyzed food items exhibit remarkably similar spectral characteristics due to their comparable elemental composition. Clearly, the significant resemblance between these spectra makes their classification challenging, at least visually. For the differentiation and classification of food samples based on their LIBS spectra, it is therefore required to employ statistical machine-learning approaches.

We chose ENET as the primary tool for analyzing LIBS spectra due to its embedded feature selection capability, which is crucial given the usage of high-resolution spectra and a restricted number of food samples [60,61]. The ENET method classifies products using LIBS while identifying the most relevant chemical constituents that support the classification results. It is important to note, however, that features identified by ML algorithms may not always represent identifiable elemental peaks and may also come from “background”. Matrix effects play a big role in how complex samples (like food) are measured by spectroscopy, and multivariate approaches may exploit the matrix effects when fingerprinting is performed [80].

To the best of our knowledge, relatively few published studies apply LIBS supported by machine-learning algorithms to discriminate/classify food samples based on their geographical origins or detection of adulteration. As for liquid food samples, three research reports have indicated that LIBS techniques paired with machine-learning approaches were employed with success for the discrimination/classification of several olive oils according to their acidity and geographical origin [54,55,56]. The olive oils tested in these studies are distinct in geographical origin and oil quality, i.e., extra virgin olive oil quality or typical commercial edible oils. Oil samples were placed in shallow, uncovered glass Petri plates such that a focused laser beam could reach their free surface to generate plasma. In these studies, classification accuracy rates of more than 90% were achieved, indicating the promise of this method. Considering the limitations and difficulties of working with aqueous samples, researchers developed a liquid-to-solid transformation of red wine using a dry collagen gel to increase the analytical performance. The LIBS technique combined with neural networks provided a classification procedure for the quality control of red wines with PDO [50]. Furthermore, the identification of milk fraud, as well as the adulteration ratios, were reported using LIBS coupled with visual clustering following principal component analysis (PCA) [29]. 

Previous studies reported using the combination of LIBS and chemometric and/or machine-learning methods to identify coffee varieties [16] and detect adulteration of wheat, corn, and chickpeas in Arabica coffee [68]. The samples were ground and pressed into pellets for LIBS measurements. Zhang et al. tested multiple classifiers (including support vector machines, neural networks, and partial least squares (PLS) regression), some of which provided an accuracy of around 80% [16]. In our study, we achieved a higher classification accuracy by employing the elastic net approach. In the other study, all major and minor elemental composition differences present in the LIBS spectra of coffee were identified using traditional chemometric techniques such as PCA and PLS [81]. In contrast, in our study, the most critical spectral features associated with elemental differences were identified using the embedded feature selection ability of the ENET model. These findings confirmed that the combination of LIBS and the ENET classifier has the potential to be used as a routine technique for determining coffee bean authenticity and detecting adulteration. It is becoming increasingly important to employ chemometrics and machine-learning methods in food authentication systems [82,83,84]. The fact that ENET allows for simultaneous feature selection (providing insights into the elemental composition), as well as classification, demonstrated that it is a method exceptionally well-suited for this food analysis task.

As far as we know, this study is the first time that LIBS and chemometric methods were used together to classify 16 types of cheese. The results showed that this combination could be a useful and practical way to find food fraud in cheese products without a lot of sample preparation. Also, this is the first study to utilize LIBS assisted by machine-learning methods to efficiently classify powdered spices using direct analysis, i.e., without making pellets. Thus, our results demonstrated that LIBS, aided by suitable statistical methods, can be an effective technique for verifying the quality and safety of spices and similar powdered products.

It is astonishing that there are discernible spectral differences between closely related cheeses. One probable explanation is that artisanal Alpine-style cheeses are produced seasonally in particular regions, and the bacteria responsible for cheese ripening and maturation are distinctively associated with geographical location and changing seasons [85,86,87,88,89].

Even though our classification experiments show a remarkably high degree of accuracy, it is important to note a critical limitation. For each example presented, the tests assume a supervised learning environment with an exhaustively defined training set. In other words, we assume that all classes are known beforehand (including the classes describing possibly fraudulent or inferior products). This cannot be guaranteed in many instances, resulting in the so-called non-exhaustive learning problem, which necessitates simultaneous class discovery and classification [90]. We plan to address this issue in future research using our prior experience with non-exhaustive training sets, such as those emerging in food safety applications [91]. 

## 5. Conclusions

The LIBS technique, paired with supervised statistical learning methods, has been evaluated in real-world applications as a rapid and robust classifier of high-value food items based on their distinctive spectral fingerprints. This study aimed to demonstrate that an existing field-deployable LIBS device originally built for material science applications may provide a rapid, easy, and inexpensive authentication platform for agricultural products where minimal or no sample preparation is required. To achieve this purpose, our study utilized new, easy, and cost-effective sample preparation techniques for liquid and powdered food samples. Utilizing nitrocellulose paper for liquid food samples improved the quality of the spectra and allowed us to avoid the typical sample splashing caused by LIBS-generated shock waves. The LIBS signal of nitrocellulose paper is readily distinguished from the spectra of tested food samples. It has also been demonstrated that accurate analysis of solid foods such as cheeses and entire coffee beans may be performed using LIBS without any sample preparation.

Overall, the results point to the feasibility of rapid identification of various high-value foods by LIBS accompanied by supervised classification methods, using not only lab-based benchtop instruments but also portable, field-deployable units. 

## Figures and Tables

**Figure 1 foods-12-00402-f001:**
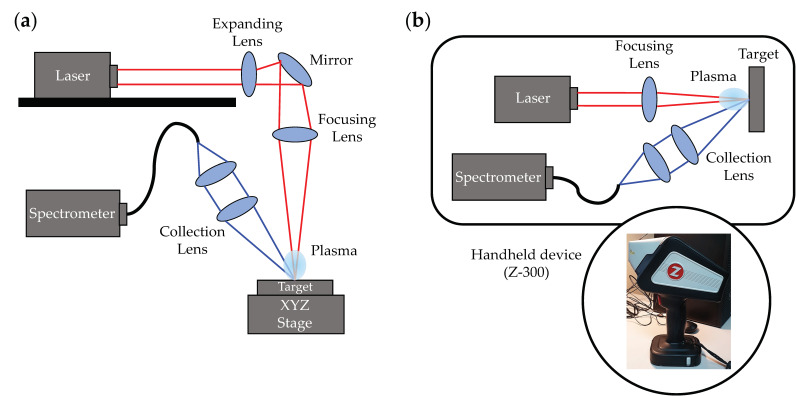
Schematic diagram of LIBS system setup; (**a**) benchtop system and (**b**) handheld system.

**Figure 2 foods-12-00402-f002:**
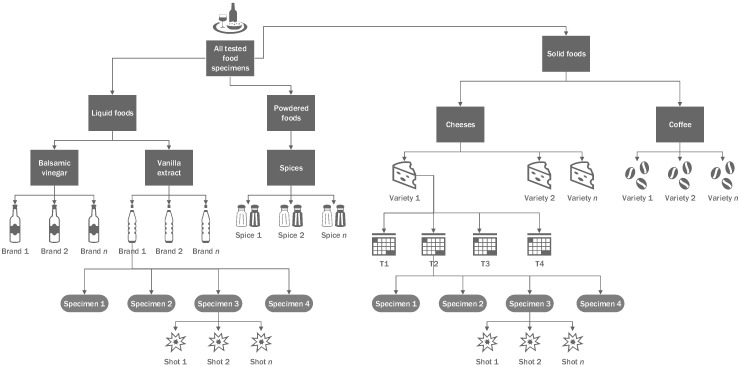
Diagram illustrating the variety of food examples and the testing procedures employed in the presented experiments. Each food product was represented by multiple specimens, each of which was interrogated repeatedly by LIBS. Please note that only cheeses were sampled at multiple time intervals.

**Figure 3 foods-12-00402-f003:**
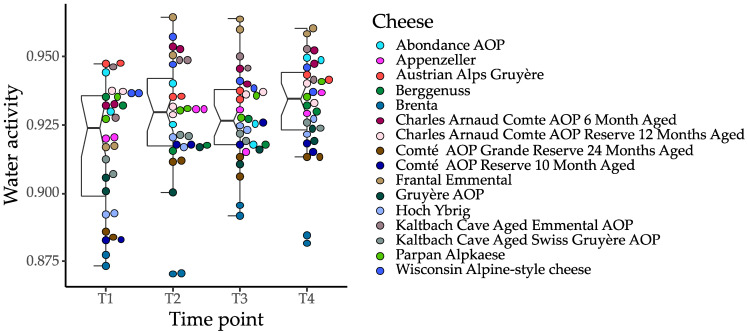
Changes in water activity in 16 types of tested cheeses over six weeks of refrigerated storage measured at four time-points.

**Figure 4 foods-12-00402-f004:**
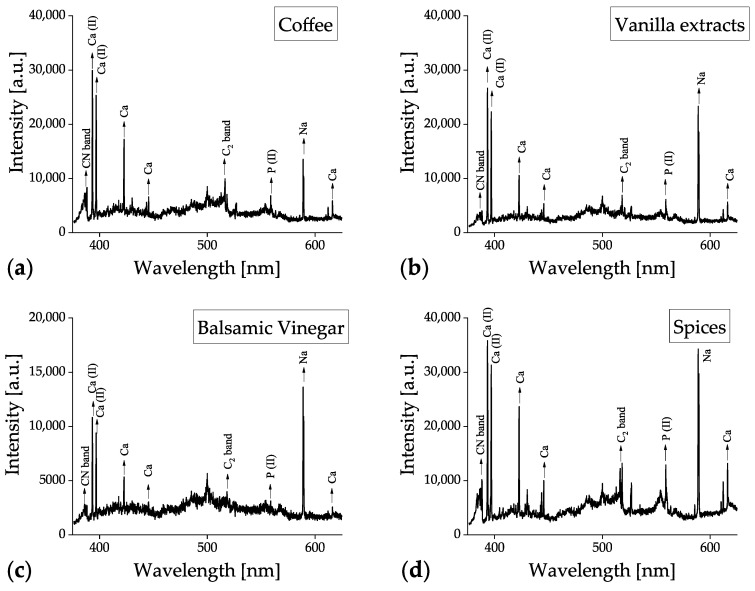
Averaged raw LIBS spectra of (**a**) coffee, (**b**) vanilla extract, (**c**) balsamic vinegar, and (**d**) spice samples collected using the benchtop LIBS system.

**Figure 5 foods-12-00402-f005:**
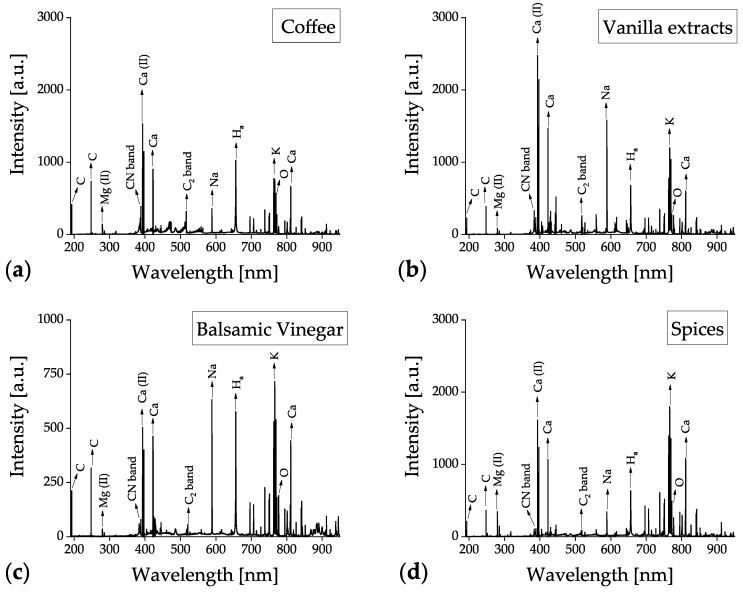
Averaged raw LIBS spectra of (**a**) coffee, (**b**) vanilla extract, (**c**) balsamic vinegar, and (**d**) spice samples collected using the handheld LIBS system.

**Figure 6 foods-12-00402-f006:**
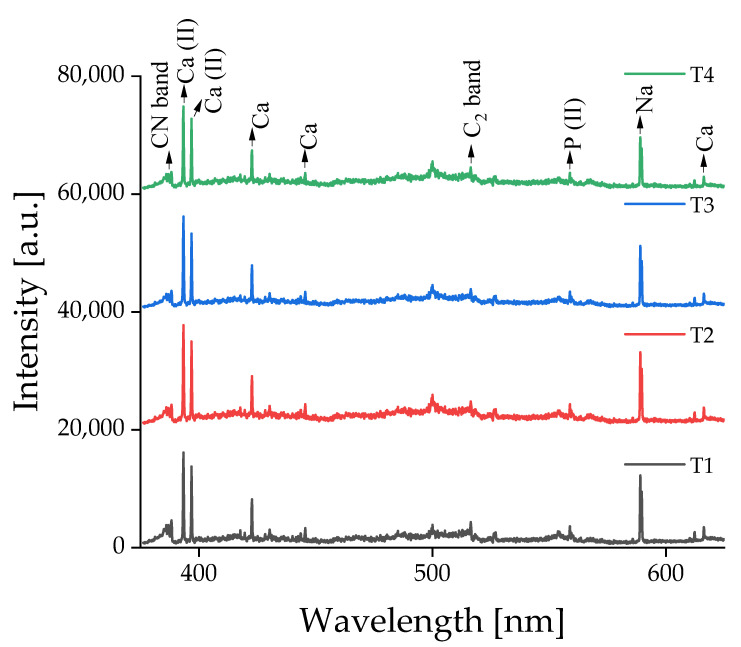
Averaged raw LIBS spectra of cheese samples measured on four different dates using the benchtop LIBS system. Note that every measurement was conducted every two weeks.

**Figure 7 foods-12-00402-f007:**
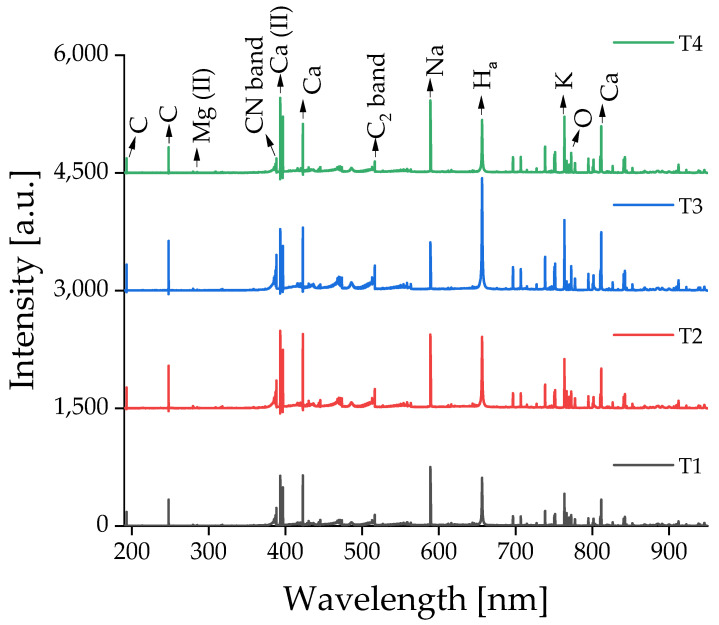
Averaged raw LIBS spectra of cheese samples measured on four different dates using the handheld LIBS system. Note that every measurement was conducted every two weeks.

**Table 1 foods-12-00402-t001:** Summary of food samples tested in the study.

Food Forms	Liquid		Solid		Powder
**Products**	Balsamic vinegar	Vanilla extract	Coffee beans	Cheeses	Spices
**Varieties or brands**	6	6	7	16	8
**Testing methods**	NC membrane	NC membrane	Surface shots	Surface shots	Surface shots

**Table 2 foods-12-00402-t002:** The averaged integrated intensity of emission peak Na I 589.0 nm at four different sampling time points in Frantal Emmental Cheese (C10). The values in parentheses represent the relative standard deviation (RSD).

Time Point	Benchtop LIBS	Handheld LIBS
T1	0.0060 (17.4%)	0.0071 (11.3%)
T2	0.0062 (10.6%)	0.0063 (15.3%)
T3	0.0056 (10.0%)	0.0067 (16.3%)
T4	0.0054 (18.6%)	0.0074 (18.3%)

**Table 3 foods-12-00402-t003:** ENET classification accuracy of five different food products measured by the benchtop and handheld LIBS systems at four different time points.

Food Products	Classifier Accuracy	
	Benchtop LIBS	Handheld LIBS
16 cheeses		
T1	85.80 ± 1.57%	81.20 ± 1.51%
T2	82.20 ± 1.53%	83.00 ± 1.34%
T3	87.60 ± 1.99%	84.70 ± 1.79%
T4	84.10 ± 1.93%	84.20 ± 1.71%
6 coffee varieties	85.00 ± 1.94%	92.70 ± 2.30%
6 vanilla extracts	94.50 ± 1.51%	98.30 ± 0.69%
6 balsamic vinegars	88.20 ± 2.10%	90.80 ± 1.88%
8 powdered spices	99.30 ± 0.70%	84.50 ± 1.94%

## Data Availability

Not applicable.

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
