# Peer review of "Rapid Food Authentication Using a Portable Laser-Induced Breakdown Spectroscopy System"

_foods, 2023, doi:10.3390/foods12020402_

Round 1
Reviewer 1 Report
This work deals with the application of benchtop and handheld LIBS systems assisted by chemometrics to classify different type of foods, while minimizing sample preanalytical processing.
The introduction and the first part is to the point and really enjoyable to read. The topic is very interesting. It is really nice to see ENET used for classification.
The manuscript however lacks clarity in many points and needs quite some work to finally highlight the value of its results.
- How many samples were used? only the sample types were mentioned but there is no info on sample number. Or were (e.g.) the 16 cheeses mentioned the actual 16 samples? If it is the latter, the phrase "The present study, as far as we are aware, for the very first time, classified 16 cheese varieties using LIBS with chemometric methods and demonstrated that this combination 434 could be a usable and practical technique to uncover food fraud in cheese products with-435 out significant sample preparation" loses value since it is to be expected that each cheese would group with itself. Please clarify
- If I understood correctly, the average spectra were used, while it would be more sound to input single replicates to build the models.
- the feature selection part deserves to be expanded, e.g. by explaining what the variables could correspond to in the various food samples. the only Figure relative to it is too small and crowded.
- the description of results, in general, should be expanded. Rather than splitting Results and discussion, it would facilitate readers to have subsections about different food samples, or at least liquid or sollid samples, and a final subsection on fraud detection.
- Figure 7 is redundant and confusing to read, The caption is very convoluted. All relevant info is already in the previous Table
All things considered it feels like the authors wanted to condensate a lot of interesting work in a single paper, but to do so they cut off some corners: clearer methods, expanded result description, and more attention towards helping the reader gain context would make this work worthy to feature in Foods.
Author Response
How many samples were used? Only the sample types were mentioned but there is no info on sample number. Or were (e.g.) the 16 cheeses mentioned the actual 16 samples? If it is the latter, the phrase "The present study, as far as we are aware, for the very first time, classified 16 cheese varieties using LIBS with chemometric methods and demonstrated that this combination could be a usable and practical technique to uncover food fraud in cheese products without significant sample preparation" loses value since it is to be expected that each cheese would group with itself. Please clarify
Thank you for the question and the chance to improve the manuscript. Indeed, our imprecise wording created confusion regarding the number of classes (categories), the number of used specimens, and the number of spectroscopic samples (i.e., the number of measurement instances). We used the word “samples” with at least three different meanings (examples, specimens, single measurement).
Generally, we used 100 measurements (samples) per food product (at one time point). In other words, a data matrix would have p columns (the number of selected spectral features) and n individual spectra (n=100×products). Every food product was represented by four physical specimens, and every one of these specimens was represented by 25 measurement samples. For instance, in the case of cheese, it would mean the following: two separate procurements (we did not mention that additional source of variability), each purchase containing 16 different blocks of artisanal cheese, each block being a source of 4 random specimens, and each specimen being sampled at 25 different locations. These measurements were repeated 4 times over the span of several weeks to ensure that variability due to storage was present in the measurements. The described procedure simulated the scenario that food inspectors would likely encounter if inspecting imported Alpine-style cheese.
For the classification procedure, 10-fold cross-validation was utilized. The dataset was divided into 10 segments of nearly equal size. In the subsequent 10 iterations of training and validation, a different fold of the data was reserved for validation, while the remaining 9 folds were utilized for learning. This entire procedure was repeated multiple times using distinct random seeds. The results of this procedure are reported as classification averages.
Obviously, we agree with an implicit critique that the procedure would be even more realistic if even more variability were represented. For instance, if the artisanal cheeses came from entirely different batches, such as different production years (to check the fingerprint stability from year to year) or were exposed to multiple different storage scenarios (proper storage vs. improper conditions). However, realistically, since the tested food products are known for small-batch production, so a recognition procedure would also operate with small batches.
Regarding the expected grouping: as we mentioned in the manuscript, the elemental composition of the food products is remarkably similar, so it is not at all evident that one could distinguish between the sampled spectra, for instance, obtained from two Alpine cheeses manufactured in two neighboring villages in Switzerland. Indeed, the results demonstrate that making these distinctions is quite difficult.
We revised some sections of the manuscript to avoid confusion. We also added Figure 2, which illustrates the sampling strategy.
We modified Table 1 to clearly indicate the number of varieties/brands of each food product (they were mistakenly identified as “samples” before).
If I understood correctly, the average spectra were used, while it would be more sound to input single replicates to build the models.
Again, it was probably a misunderstanding due to the brevity of the explanation and our imprecise language (in particular, in Table 1). The averages were shown in the Figures, but the training and classification were performed as described in response to the previous question.
The feature selection part deserves to be expanded, e.g., by explaining what the variables could correspond to in the various food samples. The only Figure relative to it is too small and crowded.
The subject of feature selection is incredibly complex, and frankly, it deserves a separate paper (which we are in the process of preparing) as it reaches well beyond the scope of this manuscript. A number of other researchers have studied feature selection in general and feature selection in spectroscopy in particular (for instance, Akulich et al., recently). Interestingly, there is no simple correspondence between selected spectral features and the actual compositional properties of foods. As explained in the paper, all foods contain similar basic chemical compounds. However, our ability to distinguish them stems from 1) subtle differences in relative composition (which are essentially “invisible” without feature selection) and machine learning and 2) the fact that some selected features clearly come from a background (the spectral regions between characteristic peaks), which are typically associated with sample structure/matrix properties. We made a short comment about this issue in the revised version.
Akulich F, Anahideh H, Sheyyab M, Ambre D. Explainable predictive modeling for limited spectral data. Chemometrics and Intelligent Laboratory Systems 225: 104572, 2022. doi: 10.1016/j.chemolab.2022.104572.
The description of results, in general, should be expanded. Rather than splitting Results and discussion, it would facilitate readers to have subsections about different food samples, or at least liquid or solid samples, and a final subsection on fraud detection.
The Results/Discussion split is part of the required journal formatting structure, but we agree that it makes following the narrative a bit difficult for this report. We tried to blur the demarcation between these two sections, to make the narration more natural and not require the reader to jump between the sections.
Figure 7 is redundant and confusing to read. The caption is very convoluted. All relevant info is already in the previous Table.
The reviewer is correct. Former Figures 7 provided a simple graphical illustration of the table, but it does not contribute any new information. Therefore, following the suggestion of two reviewers, it has been removed.
Reviewer 2 Report
The article entitled Rapid food authentication using a portable laser-induced breakdown spectroscopy system has an interesting topic, but I believe that the results of the paper might be overestimated due to the low number of the samples. In my opinion the article should be rejected and the number of samples increased.
Author Response
Whether the choice of liquid depends on the viscosity, and if so, can the specific value of the viscosity be listed for reference.
Thank you for the comments. We did not measure the viscosity of the liquid food samples. The choice of food products for the experiments was based on the report regarding food fraud, and it was not dictated by the specific physical sample properties (although we tried to cover solids, liquids, and powders). Therefore, we cannot be sure if our sample handling method would be appropriate for lower-viscosity samples (such as wine) or much higher-viscosity examples (such as olive oil). We discuss this limitation in the revised manuscript.
Choose one of them in Table 3 and Figure 7, both of which have the same meaning, saving space in the publication space.
Per recommendation from two reviewers, we removed Figure 7, which provides only a graphical representation of Table 3, and does not contribute any additional information.
What is the basis for selecting the variety of raw materials?
The key motivation for choosing particular food products were publications, news, and industry reports of food fraud cases, mainly in the USA and EU. More specifically, we were interested in small-batch food products rather than mass-produced ones. We used food fraud databases listed in articles such as Manning L, Soon JM. Food fraud vulnerability assessment: Reliable data sources and effective assessment approaches. Trends in Food Science & Technology 91: 159–168, 2019. doi: 10.1016/j.tifs.2019.07.007.
I don't understand why the spectral range of handheld devices is higher than that of desktop devices, is it a problem with the device itself or something else?
Typically, laboratory benchtop devices are optimized for sensitivity and spectral resolution but not necessarily spectral range. That was the case with our instrument. However, the handheld device, which was built for alloy testing, was mostly optimized for spectral range in order to accommodate a variety of metals. However, given that the alloy sampling is hardly ever limited by the sample size, and typically the operators enjoy the overabundance of samples (for instance, in scrap metal testing), the sensitivity of these devices is much lower. The different tasks and objectives result in different engineering and optimization techniques. One of our research goals was to check whether a commercially-available device optimized for a dramatically different research task (alloy testing) may be helpful in practical food testing scenarios. We were pleasantly surprised to learn that, indeed, it can, and in fact, it performs pretty well for specific food groups.
At present, whether the handheld devices mentioned in this article have been put into production practice or only remain in the theoretical stage.
As mentioned above and stated in the paper, the handheld device is a commercial system designed and optimized for alloy testing. It is available for use. However, it comes with a built-in classification engine that is optimized for its original task. In our continued collaboration with the manufacturer, we hope to create a self-contained version in which the on-board classifier operated by a portable microcomputer would work for food classification.
If the selected raw materials are affected by weather, season, and other factors during the growth process, can the model achieve the expected results? For example, whether the same coffee beans picked in different years can be identified. Due to the consideration of portable equipment into practice later.
This is one question to which we have no answer, sadly. The seasonal character of some products may indeed be the reason for our ability to classify them. For instance, it is known that the highest-quality, rarest Alpine cheeses in Gruyere style, such as L’Etivaz AOP, are manufactured seasonally, only in the summer months. So, it is quite probable that our ability to distinguish very similar cheeses, manufactured supposedly according to a similar process, is at least partially driven by the fact that bacterial cultures and milk composition are affected by the seasons. There is some evidence in the literature suggesting that this may be a plausible hypothesis:
- Barron LJR, Fernández de Labastida E, Perea S, Chávarri F, de Vega C, Soledad Vicente M, Isabel Torres M, Isabel Nájera A, Virto M, Santisteban A, Pérez-Elortondo FJ, Albisu M, Salmerón J, Mendı́a C, Torre P, Clemente Ibáñez F, de Renobales M. Seasonal changes in the composition of bulk raw ewe’s milk used for Idiazabal cheese manufacture. International Dairy Journal 11: 771–778, 2001. doi: 1016/S0958-6946(01)00120-0.
- Montel M-C, Buchin S, Mallet A, Delbes-Paus C, Vuitton DA, Desmasures N, Berthier F. Traditional cheeses: Rich and diverse microbiota with associated benefits. International Journal of Food Microbiology 177: 136–154, 2014. doi: 1016/j.ijfoodmicro.2014.02.019.
- Van Hekken DL, Drake MA, Tunick MH, Guerrero VM, Molina-Corral FJ, Gardea AA. Effect of pasteurization and season on the sensorial and rheological traits of Mexican Chihuahua cheese. Dairy Sci Technol 88: 525–536, 2008. doi: 1051/dst:2008016.
- Sánchez-Gamboa C, Hicks-Pérez L, Gutiérrez-Méndez N, Heredia N, García S, Nevárez-Moorillón GV. Seasonal influence on the microbial profile of Chihuahua cheese manufactured from raw milk. International Journal of Dairy Technology 71: 81–89, 2018. doi: 1111/1471-0307.12423.
We don’t know whether this is true for coffee, as the coffee producers do not typically provide detailed information about the time of the harvest. Of course, we know that coffee is generally harvested from September to March (north of the equator) and from April to August (south of the equator). These are intriguing questions, and we mentioned them in the revised discussion.
Reviewer 3 Report
This research deployed commercial handheld LIBS equipment to development a novel portable methods for detection. It focused on regional agricultural commodities such as European Alpine-style cheeses, coffee, spices, balsamic vinegar, and vanilla extracts. The pre-processed and standardized LIBS spectra were used to train and test the elastic net-regularized multinomial classifier. In my opinion, it must be improved. Thus, I recommend major revision according to the following comments.
1. Whether the choice of liquid depends on the viscosity, and if so, can the specific value of the viscosity be listed for reference.
2. Choose one of them in Table 3 and Figure 7, both of which have the same meaning, saving space in the publication space.
3. What is the basis for selecting the variety of raw materials?
4. I don't understand why the spectral range of handheld devices is higher than that of desktop devices, is it a problem with the device itself or something else?
5. At present, whether the handheld devices mentioned in this article have been put into production practice or only remain in the theoretical stage.
6. If the selected raw materials are affected by weather, season and other factors during the growth process, can the model achieve the expected results? For example, whether the same coffee beans picked in different years can be identified. Due to the consideration of portable equipment into practice later.
Author Response
In this paper, the authors propose the application of technique LIBS to authenticate various products. The number of samples is very small for each product and does not allow any differentiation.
We believe that the reviewer was misled by our imprecise wording and the use of the term "sample" to convey at least in three different meanings (food examples, food specimens, and single measurement). See the detailed explanations provided to Reviewer 1. What we referred to in Table 1 and the rest of the manuscript as “samples” were really varieties or examples of the food products (we corrected the language). Each product was represented by multiple specimens and 100 spectroscopic samples/measurements from multiple sites.
The small-batch food products, such as regional artisanal cheeses, do not allow for a massive number of representative exemplars because the product changes from year to year or vintage to vintage (wine, artisanal cheese, or balsamic vinegar are examples of such products). Therefore, we tried to simulate the variability in the measurements that a food inspector may execute by performing multiple independent measurements, measuring multiple independent locations at the specimen, and repeating the measurement over a span of time to account for natural variability and (as in the case of cheeses) different storage period.
Remarks – Corrections
In Table 1 lists 7 samples of balsamic vinegar while line 98 lists six. There is confusion.
Corrected
Line 110. The term DI is mentioned for the first time. Therefore Deionized (DI)
Corrected
Chapter 2.1.1. Were the balsamic vinegar samples dried or not?
Ten microliters of balsamic vinegar were deposited onto a 6×6-mm square of nitrocellulose membrane and dried at room temperature for 30 minutes. We also revised accordingly in the text.
4. Line 125. Not ten microliters but 10 μL.
Corrected
Reviewer 4 Report
In this paper, the authors propose the application of technique LIBS to authenticate various products.
The number of samples is very small for each product and does not allow any differentiation.
Remarks – Corrections
1. In Table 1 lists 7 samples of balsamic vinegar while line 98 lists six. There is confusion.
2. Line 110. The term DI is mentioned for the first time. Therefore Deionized (DI)
3. Chapter 2.1.1. Were the balsamic vinegar samples dried or not?
4. Line 125. Not ten microliters but 10 μL.
Author Response
In Table 1 lists 7 samples of balsamic vinegar while line 98 lists six. There is confusion.
Corrected
Line 110. The term DI is mentioned for the first time. Therefore Deionized (DI)
Corrected
Chapter 2.1.1. Were the balsamic vinegar samples dried or not?
Ten microliters of balsamic vinegar were deposited onto a 6×6-mm square of nitrocellulose membrane and dried at room temperature for 30 minutes. We also revised accordingly in the text.
Line 125. Not ten microliters but 10 μL.
Corrected
Round 2
Reviewer 1 Report
The paper improved greatly and can be published as is.
We revised some sections of the manuscript to avoid confusion. We also added Figure 2, which illustrates the sampling strategy.
Figure 2 perfectly addresses all concerns
Obviously, we agree with an implicit critique that the procedure would be even more realistic if even more variability were represented
This will always be said for each and every submitted manuscript in the history of chemometrics. But I believe the paper is quite complete now.
Reviewer 2 Report
The article entitled Rapid food authentication using a portable laser-induced breakdown spectroscopy system has an interesting topic, but I believe that the results of the paper might be overestimated due to the low number of the samples. In my opinion the article should be rejected and the number of samples increased.
Reviewer 3 Report
no further comments.
Reviewer 4 Report
I consider the revised version of the paper to be satisfactory and publishable